# Non-Standard Technical Solutions in Polish Tie-Stall and Loose-Housing Barns: Farmer Initiatives to Improve the Comfort of Dairy Cattle

**DOI:** 10.3390/ani16010064

**Published:** 2025-12-25

**Authors:** Marek Gaworski, Michał Boćkowski

**Affiliations:** 1Department of Production Engineering, Institute of Mechanical Engineering, Warsaw University of Life Sciences, 02-787 Warsaw, Poland; 2Metal Structures Company, 07-300 Ostrów Mazowiecka, Poland; michalbockowski@tlen.pl

**Keywords:** barn, cattle, comfort, technical solutions, welfare

## Abstract

Technical barn equipment and its associated impacts on cattle welfare are subjects of scientific evaluation. In practice, however, dairy cattle housing conditions are also assessed directly by the farmers responsible for the animals’ living conditions. Therefore, in this study, farmers were asked about their approaches to improving the housing conditions and comfort of dairy cattle in their barns. During visits to 40 dairy farms, the farmers’ implementation of non-standard technical solutions in tie-stall and loose-housing barns was assessed. Thirty-two farms employed at least one non-standard technical solution. The number of technical ideas for improving cattle comfort was highest in barns with tie-stall systems. In barns with this housing system, the greatest number of non-standard technical solutions was found in feeding areas, while in loose-housing barns, the majority were found in social areas. This study was descriptive and was not intended to directly measure animal comfort or welfare outcomes; instead, it entailed documenting the solutions implemented by farmers and their perceived effects.

## 1. Introduction

The efficiency of cattle production is the result of a combination of many factors, including breed improvement and selection, rational feeding, and the creation of appropriate climatic and microclimatic conditions for animal husbandry, especially within livestock buildings [1]. A significant problem that requires intensive research and experimental work is determining the impact of the barn environment on animal productivity [2], particularly the effects of climatic conditions and the building’s technical infrastructure on livestock organisms; achieving optimal environmental conditions in animal housing involves the use of technical and structural solutions for buildings intended for dairy cattle production [3].

Dairy production on a farm is a complex and highly labor-intensive process [4]. On the one hand, it is essential to provide animals with appropriate environmental conditions in order to achieve the highest possible production results; on the other hand, the technology and organization of animal production should consider the reduced labor intensity of animal handling [5] in areas such as feeding, milking, and manure removal, while also creating the best possible living conditions for the animals. To reduce the labor intensity of animal handling, the functional layout of cattle buildings should include simple technological and communication routes, especially for feed and manure handling [6].

In cattle production, to improve economic results by increasing efficiency and reducing production costs, it is also necessary to consider animal welfare and environmental protection. Technology in livestock breeding is one of the key production factors that influences animal welfare [7].

The technical and technological solutions farmers use in their livestock buildings have fundamental impacts on meeting welfare criteria. The dimensions of lying stalls [8], the temperature and humidity in the barn [9], the type of floor [10], and the equipment used for manure removal [11] all impact an animal’s welfare, influencing its proper development and productivity. Particular attention to the welfare of farm animals focuses on the way they are housed on large industrial farms [12], as these systems often deprive animals of the ability to satisfy their basic behavioral and physiological needs, which is a source of suffering.

Providing proper animal welfare depends largely on technical factors. Dairy farming has made enormous technological progress in recent years [13], driven by the need to mechanize arduous manual livestock work [14] and to ensure an appropriate level of animal welfare. The appropriate design and equipment of individual technological areas within a building can improve herd welfare [15]. Improper use of technology is one of the main causes of deterioration in living conditions, which impacts the welfare of cattle in the herd [16]. Therefore, farmers’ awareness of the recommendations and technical standards in various areas of the barn remains important, translating into motivation to adapt the bedding, flooring, manure removal systems, and other technical elements of the barn equipment to the actual needs of the animals [17], and, if necessary, to introduce modifications to the technical equipment in areas where animals come into contact [18]; an example of modernization implemented by farmers is covering parts of the barn floors, where cows most often walk and stand, with special rubber mats [19]. Farmers are also asked about these modifications, or modernizations, when assessing experiences related to improving technological operations undertaken in barns [20], which confirms the importance attached to farmers’ opinions regarding the effects of their activities on dairy farms; the effects of these activities on improving cow comfort can be supported with an appropriate assessment tool [21], which identifies critical areas of dairy cattle comfort, taking into account the design of the lying areas and their available space, technical solutions in the feeding area, and the design of passageways.

Discussions regarding technology and technical equipment in livestock facilities and cattle welfare indicate the growing importance placed on assessing sources of animal comfort. Naturally, these sources of comfort are the subjects of detailed assessments in numerous scientific studies. In practice, dairy cattle housing conditions, which can impact the animals’ comfort and welfare, are also assessed directly by farmers; after all, farmers are directly responsible for the conditions in which animals are housed in livestock buildings, so these conditions can, and even should, be assessed by farmers. Therefore, it may be worthwhile to ask farmers about their approaches to improving the housing and comfort of dairy cattle in barns. It is important to note, however, that farmers’ perceptions of “comfort” may differ from scientific indicators of animal welfare. Therefore, in this research study, we focus on documenting farmers’ technical proposals and responses, rather than confirming and validating welfare outcomes.

Considering the presented spectrum of considerations, it is possible to formulate the following research problem: to identify the actions farmers believe improve cattle comfort in livestock housing. In this case, farmers’ actions can be understood (interpreted) as the implementation of practical ideas for improving the technical equipment of barns in order to increase the comfort of cattle.

The aim of this study was to assess the implementation of non-standard technical solutions in barns and compare them based on animal housing system criteria. The cognitive goal of this study was to develop an index that would provide a basis for comparing the implementation of non-standard technical solutions in barns and other similar livestock facilities. The utilitarian goal of this study was to gather knowledge about non-standard technical solutions in barns that would be useful for consideration and evaluation by stakeholders.

The research undertaken, and its objectives, stem from the following gap in the research area: despite substantial work on standard barn designs and commercial equipment, farmer-initiated, “non-standard” modifications are rarely described in a systematic way.

The term “non-standard solution” was understood in this research and article to refer to unconventional technical solutions used in livestock buildings by farmers for a specific purpose. In practice, farmers have ideas for improving cattle comfort by introducing non-standard technical solutions, such as self-designed stall partitions, alternative barriers in the feeding area, or customized water access points.

The key steps of this research study are developed in the following chapter, regarding materials and research methods.

## 2. Materials and Methods

The research objects were farms and their barns, located in the Mazowieckie and Podlaskie Voivodeships, i.e., regions with favorable conditions for dairy production in Poland.

### 2.1. General Characteristics of the Visited Livestock Facilities

In total, the design of the research study allowed us to visit 40 dairy farms. The group of farms included in this study was a purposeful sample of farms where, based on previous contacts and knowledge, non-standard technical solutions were expected in barns. Each farm had one barn for dairy cattle, where observations and evaluation of non-standard technical solutions could be conducted. The group of 40 barns included facilities representing two basic animal housing systems: tie-stall (with limited freedom of movement) and loose-housing systems. The structures of the studied facilities included 19 farms with tie-stall barns and 21 farms with loose-housing barns; the second group comprised 19 barns with a free-stall housing system and two barns with a common lying area.

Farms with tie-stall barns had herds of cows ranging from 10 to 65 animals, while farms with loose housing had herds ranging from 35 to 210 cows.

In the barns with tie-stall housing systems, two types of bedding were used: rubber mats on 1 farm, and straw (shallow bedding) on 18 farms. In the barns with loose-housing systems, three types of bedding were identified: rubber mats (on 9 farms), low straw bedding (on 10 farms), and deep straw bedding in 2 barns with common lying areas.

On farms with tie-stall barns, cows were milked using bucket milking systems (6 farms) and pipeline milking systems (13 farms). On farms with loose housing, cows were milked in milking parlors (18 farms) and with milking robots (3 farms).

A summary of data characterizing the dairy farms included in this study is presented in Table 1.

Each type of bedding surface has its own unique characteristics. If farmers were to implement non-standard technical solutions specifically related to the bedding surface, the specific characteristics of these surfaces could influence the likelihood of implementing non-standard technical solutions. Straw bedding dominated in barns with tie-stall systems compared to loose-housing barns. Barns with deep straw bedding may offer more flexibility in implementing non-standard technical solutions.

### 2.2. Approach to Research Data Collection

Due to prior collaboration and professional responsibilities, one of the authors had contact with numerous dairy farms, which enabled the preliminary selection of farms with barns where non-standard technical solutions in cattle housing could be expected. During the design phase of the research study, the owners of 65 dairy farms were initially contacted for selection in this study. A group of 21 farmers declined the opportunity for us to visit the farm and conduct the research; a further 4 farmers were undecided and, therefore, not further considered. Prior evidence that at least one non-standard solution could be identified was not required for inclusion in this study.

Efforts were made to obtain consent from farmers to visit their farms to conduct observations. The purpose, scope, and details of the planned research were presented to farmers via telephone or in person, providing important arguments in the discussion leading to the farmers’ consent to visit their farms and barns.

The research visit to each farm was preceded by a phone call to schedule the visit. All observations of non-standard technical solutions in the barns and discussions with farmers regarding these solutions were conducted within a single day. A typical visit to a dairy farm lasted approximately three hours. Only one farm was visited per day. Visits were conducted during the spring and summer months, scheduled around noon. One of the co-authors, who works professionally with farms, participated in all farm visits. The other co-author participated in the research conducted on 34 dairy farms.

During the visit to each farm, the areas related to the animals’ housing in the barn and its surroundings were inspected; the surroundings of 12 barns included outdoor runs (paddocks) for cattle, directly connected to the livestock building; the inspection aimed to identify non-standard technical solutions in the cattle facilities.

The methodological approach to this research included identifying non-standard technical solutions independently in the following four zones/areas of the barns: feeding, lying, milking, and social areas. Feeding, lying, and milking areas clearly identify their places within the barn structure. We also included manure alleys and manure removal infrastructure within the social zone; additionally, outdoor runs (paddocks) and equipment such as brushes were also included as part of the social area. For complete information, the studies/observations in the feeding zone also considered an assessment of the animals’ access to water. For this research, we focused on the locations (with identifiable non-standard technical solutions) where all groups of dairy cattle were housed on farms, including dairy cows, calves, heifers, and dry cows.

During each barn inspection, detailed information about identified non-standard technical solutions was recorded on a research form, a type of protocol dedicated to the needs of research (including observations) carried out on dairy farms, which included a checklist for assessing key elements of individual zones in the barn. A simplified diagram of the solution was also created on the same form. For complete documentation, photos of non-standard technical solutions were taken in individual zones (areas) of the visited barns. During the barn inspection, no formal coding scheme was used for non-standard solutions. A decision rule was adopted, defining a non-standard solution as a technical structure that constitutes a complementary element within the regular structure of the building, with features that modify the existing functionality at the location of the new element.

Individual non-standard technical solutions were discussed with farmers during the inspection. The purpose of the discussions was to learn farmers’ opinions on the solutions, what inspired their design, and how they might impact animal comfort, behavior, preferences, and other aspects. The most important issues raised by farmers were recorded on a working form in a manner that allowed for the extraction of key information for further data analysis, specifically the assessment of the expected changes in animal comfort resulting from the use of non-standard technical solutions.

For additional calculations in this research study, the areas designated for the herd in the barn were measured. Two types of equipment were used for the measurements: a tape measure (with a measuring range of up to 7.5 m and a measurement accuracy of 1 mm) and a Würth laser meter (with a measuring range of up to 200 m and a measurement accuracy of 1.5 mm) with an optical reference point system. The measurement accuracies, given in [mm], were not relevant to the measurements performed; they only characterized the measuring devices.

The measurements of usable floor space in the barns were justified through a planned, additional calculation, which involved our proposal of calculating the index of non-standard solutions in livestock buildings (S_ns_) based on the following formula:(1)$$S_{ns}\;=\;\frac{N_{ns}}{A_{u}}$$
where S_ns_—index of non-standard solutions in livestock buildings [1/m^2^]; N_ns_—number of non-standard solutions in the barn [-]; and A_u_—usable floor space in the barn [m^2^].

In Equation (1), the area designated for cattle is considered the area of the livestock building where various animal activities take place, including four zones within the livestock building (bedding, social, feeding, and milking). The usable area (usable floor space in the barn) was interpreted in this study as the sum of the areas available for cows and other areas where non-standard solutions could be identified, i.e., passageways and calf pens. The area for animals is considered for both tie-stall and loose-housing systems. The calculation formula for the index of non-standard solutions in livestock buildings included (in the denominator) the usable floor space in the barn. The choice of this area was justified by its constant value in the livestock facility. An alternative analytical approach could have been to relate the number of non-standard technical solutions to the number of cows or dairy cattle in the livestock facilities; however, the herd population on a dairy farm is not a constant, so calculating the index of non-standard solutions in livestock buildings would involve determining only its instantaneous value. Other reference points in the calculation of the S_ns_ index could include the number of lying stalls or the number of pens in the barn; however, these data do not allow for full comparisons of different livestock facilities if they incorporate loose-housing systems without dedicated lying stalls.

The proposed index of non-standard solutions in livestock buildings, S_ns_, implicitly treats all non-standard technical solutions in the barn as equivalent, regardless of their complexity or assumed impact.

### 2.3. Statistical Analysis

Statistical analysis of the collected data was performed using Statistica v.13.3 software [22]. The descriptive statistical indicators, i.e., mean and median, were determined for the assessed S_ns_ indices in farms with two kinds of housing systems. The comparison of the S_ns_ indices in dairy farms with tie-stall and loose-housing systems was carried out using the Mann–Whitney U-test, with a significance level of 0.05.

## 3. Results

While collecting research material directly in livestock facilities, attention was paid to the varying intensity of occurrence for various types of non-standard technical solutions in the areas where animals live.

The results of this study (inspection of non-standard solutions), along with other details in the tie-stall barns, are presented in Table 2; these data come from 19 barns with tie-stall housing systems. At least one non-standard technical solution was identified in each barn with a tie-stall system; in total, 22 such solutions were found in this group of barns. The largest number of non-standard technical solutions was found in feeding areas, with 15 instances, followed by the lying and social areas, with 6 and 1 instances, respectively. No non-standard solutions were identified that could be clearly assigned to milking cows in the tie-stall barns. Although the milking areas overlap somewhat with the lying areas in tie-stall barns, the non-standard technical solutions, due to their intended uses, could be clearly assigned to the lying area.

The results of this study (inspection of non-standard solutions), along with other details in the loose-housing barns, are presented in Table 3, coming from 21 barns with loose-housing systems.

Farmers had certain expectations regarding the effects of the implemented solutions in their barns, listed in the penultimate columns of Table 2 and Table 3. Farmers responded to these expected effects by stating whether they were achieved or not; a “Yes” response indicated that the intended effect was achieved, while a “No” response indicated that the expected effect of implementing a given technical solution in the barn was not achieved. The distribution of these responses (“Yes” and “No”) is presented in the last columns of Table 2 and Table 3.

Photos illustrating examples of non-standard technical solutions in the visited dairy cattle farms are presented in Figure 1 and Figure 2.

After identifying a non-standard solution for the barn, a crucial step was to determine the source of inspiration for the designer; when implementing a technical non-standard solution in the barn, farmers were often guided by the following considerations: reducing workload, reducing feed/water losses, and reducing the use of bedding material. Moreover, farmers pointed to another set of reasons, namely, improving animal safety, enhancing living conditions, and enhancing hygiene, all of which are combined with comfort and welfare; that is, half of the listed reasons directly relate to improving animal welfare, while the remaining three focus on the economics of dairy production (feed/water and bedding losses) and labor costs.

To summarize the categories of farmers’ motivations/inspirations for implementing non-standard technical solutions in different zones of tie-stall barns, based on the data in Table 2, the most frequently mentioned motivations were improving hygiene and comfort (20 responses) and reducing workload (19 responses), while the fewest responses (2 cases) concerned improving living conditions. To summarize the categories of farmers’ motivations/inspirations for implementing non-standard technical solutions in different zones of loose-housing barns, based on the data in Table 3, the most frequently mentioned motivations were reducing workload (15 responses) and improving animal safety (10 responses); the fewest responses (3 cases) concerned reducing the use of bedding material.

Of the 40 farms surveyed, 32 facilities identified individual technical solutions that the farmers had implemented. Of these, 26 facilities identified one non-standard solution, and 6 facilities identified two solutions, including 2 farms with two non-standard solutions in one (feeding) zone; in 8 farms (out of 40 surveyed), all with loose-housing barns, no non-standard technical solutions were found.

In general, the distribution of non-standard technical solutions in the studied areas of all barns was as follows: 10 farms had non-standard solutions in the lying area (including 6 farms with tie-stall system and 4 farms with loose-housing systems), 20 farms had non-standard solutions in the feeding area (including 15 farms with tie-stall systems and 5 farms with loose-housing systems), and 8 farms had non-standard solutions in the social area (including 1 farm with a tie-stall system and 7 farms with loose-housing systems). No non-standard solutions were found in the milking areas of any of the studied facilities (barns).

In the tie-stall barns, non-standard solutions dominated the feeding area; a characteristic finding was the observation of the same non-standard solution in as many as 10 farms out of 19 tie-stall barns.

In the barns with loose-housing systems, the majority of farms had non-standard solutions in the social zone (7 farms out of 21 farms with loose-housing systems); the zone with the smallest number (4) of identified non-standard solutions in the surveyed farms was the lying zone.

The general comparison of data in Table 2 and Table 3 reveals that the majority of non-standard solutions are associated with farms using tie-stall housing systems.

Considering all cases of non-standard solutions identified in tie-stall barns, we calculated the percentage share of non-standard solutions in each area. The results of this calculation are presented in Figure 3a. Figure 3b presents the results of a similar calculation based on data from loose-stall barns.

Farmers assessed the effects of the implemented non-standard solutions in their barns based on six rating categories: improving animal safety, improving hygiene and comfort, improving living conditions, reducing feed/water losses, reducing the use of bedding material, and reducing workload. Farmers reported whether the expectations associated with the implementation of non-standard solutions in the barns were met, as either “Yes” (expectation met) or “No” (expectation not met); this information was used to calculate the percentage of positive ratings out of all ratings (positive: “Yes” and negative: “No”) for each of the six rating categories; these calculations were performed separately for tie-stall and loose-housing barns and are presented in Figure 4a and Figure 4b, respectively. The total number of “effect evaluations” (Yes/No) for each category in tie-stall barns was as follows: improving animal safety (7 farmer ratings), improving hygiene and comfort (20), improving living conditions (2), reducing feed/water losses (16), reducing the use of bedding material (4), and reducing workload (19). The total number of “effect evaluations” (Yes/No) for each category in loose-housing barns was as follows: improving animal safety (10 farmers’ ratings), improving hygiene and comfort (8), improving living conditions (7), reducing feed/water losses (5), reducing the use of bedding material (3), and reducing workload (15).

Based on data collected on the farms, the index of non-standard solutions in livestock buildings (S_ns_, according to Formula (1)) was calculated for each dairy farm. The values of the index of non-standard solutions in livestock buildings with tie-stall housing systems ranged from 0.0012 to 0.0192 [1/m^2^]. The values of the index of non-standard solutions in livestock buildings with loose-housing systems ranged from 0.00 to 0.0023 [1/m^2^]. An S_ns_ value of 0.00 indicates that no non-standard solutions were identified in the visited barns. The mean values of the index of non-standard solution (S_ns_) for tie-stall and loose-housing barns were 0.0042 [1/m^2^] and 0.0008 [1/m^2^], respectively, while the medians were 0.0031 [1/m^2^] and 0.0007 [1/m^2^], respectively. The results of the comparison of the two groups of barns using the non-parametric Mann–Whitney U test indicated a statistically significant difference (*p* < 0.001) in the index of non-standard solutions for tie-stall and loose-housing barns.

## 4. Discussion

Summarizing the research we conducted on 40 dairy farms in Poland, it turned out that 80% of them had at least one non-standard technical solution in the barn; these 32 farms included all 19 farms with tie-stall systems for housing dairy cattle; in barns with this housing system, most non-standard technical solutions were found in the feeding area, while, in barns with loose-housing systems, most of them were in the social area. The proposed index of non-standard solutions in livestock buildings (S_ns_), identifying the number of such solutions in relation to the usable floor space in the barn, was higher in barns with tie-stall housing systems (average S_ns_ 0.0042 [1/m^2^]) compared to loose-housing barns (average S_ns_ 0.0008 [1/m^2^]).

As a result of visits to dairy farms and observations, the following research objective was achieved: identifying non-standard technical solutions in barns, proposed and implemented by farmers; these non-standard technical solutions, as confirmed with the presented research, are applicable to both tie-stall and loose-housing barns; this, in turn, can serve as a premise justifying the comparison of groups of livestock facilities with tie-stall and loose-housing systems. Non-standard technical solutions can be another practical criterion for comparing barns with the two identified cattle housing systems. The above comparison criteria are numerous in practice, and the research undertaken aims to identify the benefits that justify the development of a given cattle housing system.

One of the key criteria for comparing tie-stall and loose cattle housing systems is animal welfare, particularly when considered from the perspective of its improvement [23]. One study [24], following the guidelines of the Welfare Quality protocol, demonstrated significant differences between tie-stall and loose housing for dairy cows. The loose-housing system, particularly the free-stall system, is more beneficial for cattle in terms of nutrition, housing conditions, and the ability to exhibit natural behaviors. Observations have confirmed that free-stall housing encourages cows to exhibit their natural behaviors in various areas of the barn. For example, in the lying area, the loose-housing system allows cows to express preferences regarding the quality of the lying surface [25]. When housing cows in free-stall pens, it has been demonstrated that cows can express preferences regarding the location of lying stalls between [26] and within the rows [27]. While the tie-stall system does not allow for the identification of many behaviors and preferences, it remains a significant alternative to the loose-housing system, particularly when comparing certain aspects of animal life processes. In exemplary studies [28], feeding and rumination behaviors, as well as stress signals, were assessed in cattle housed in tie-stall and free-stall systems. It was found that tied animals were more stressed than those kept in free stalls. Comparison of various indicators for dairy cattle in alternative housing systems can be used to formulate practical recommendations for farmers; for example, based on evaluation studies of the production and reproductive performance of cross-bred dairy cattle [29], the following recommendations were made for one region in India: a loose-housing system for housing dairy cattle during summer.

In addition to assessing various aspects of cow nutrition, comfort, and behavior, the comparison of tie-stall and loose-housing systems also considers other criteria that constitute arguments in the discussion on the development of dairy production technologies on dairy farms. One of the important comparative factors is the number of cases of mastitis found in herds kept in tie-stall and free-stall systems, which was further explored in a Swiss study [30]; it turns out that the frequency of mastitis is related to management practices typical of both barn-based cattle systems. Another characteristic of cows in a herd, taken into account in comparisons between tie-stall and loose-housing barns, is the BCS (Body Condition Scoring) index. In one study [31], it was found that the distribution of body condition scores for cows with different daily yields, at different ages, and assessed in different seasons was significantly related to the housing system; cows kept in tie-stall barns were more likely to receive extreme body condition scores (no more than 2.0 or 4.0 and more BCS points) compared to cows in loose-housing barns. While developing comparative studies of animals kept under two systems (tie-stall and free-stall), attention was also paid to udder health and milk composition [32]; when comparing two systems of housing cows on one farm, it was found that milk production under the free-stall system was associated with a higher somatic cell count and a less favorable milk composition and fatty acid profile. The importance of the cow housing system in the context of milk quality is confirmed by the use of raw milk for the production of specialized dairy products; studies on the quality of milk from cows under two housing systems were carried out in the Parmigiano-Reggiano cheese production region [33] and indicated the additional need to take climatic conditions into account in connection with management strategies for the analysis of dairy production.

Including alternative dairy cattle housing systems in research generates a wide range of comparisons, encompassing not only the animals but also the technical conditions of the livestock buildings [34] and modifications [35]. However, even a comparison of the technical conditions of barns with tie-stall and loose-housing systems ultimately boils down to the question of cattle welfare in the studied livestock facilities; therefore, animal welfare is a key criterion for comparative assessments of tie-stall and loose-housing systems in systematic reviews [36].

The research presented in this article highlights the crucial role of farmers in improving the housing conditions of dairy cattle in livestock buildings, which aligns with research underscoring the importance of farmers’ perceptions of improved animal welfare, as well as their sensitivity to farm animal welfare (FAW), which translates into the need for subsidies that support structural changes [37]. Of course, the individual personalities of farmers and their attitudes towards the importance of animal welfare [38], including dairy cattle, as well as their awareness of the risks associated with decisions made regarding animal health, welfare, productivity, and farm management, are important [39]. Farmers’ perceptions and personalities were not directly assessed in this study, but they provide interpretative context and a source of guidance for future research, such as examining the links between farmers’ attitudes and their willingness to implement non-standard solutions in dairy production.

Farmers’ awareness may be one of the key factors in explaining the results of our own study, which focused on equipping two types of barns with non-standard technical solutions. Owners of tie-stall barns are aware of the imperfections of a housing system in which cows are tethered and have limited freedom of movement [40], which in turn, may prompt farmers to take more intensive activities, increasing the need to adapt or modify the environment to provide cows with greater comfort and reduce stress [41], a thesis that was supported by the study presented, which identified non-standard technical solutions on all farms visited with tie-stall housing systems. In comparison, only 62% of visited farms with loose-housing systems used non-standard technical solutions.

Cows in tie-stall barns, regardless of their daily lying time (10–12 h/day) [42], spend a significant portion of their lives in one place, i.e., in a lying area connected to a feeding area (feed alley, manger); therefore, dairy herd owners may intuitively pay more attention to these areas (lying and feeding) in the barn to ensure that the cows achieve a high level of comfort. To confirm this, in the presented study, the most non-standard solutions introduced in tie-stall barns by farmers were found in the feeding area, followed by the lying area (Figure 3a). Farmers independently improve lying and feeding areas for dairy cattle; their actions may support the implementation of modernized designs for technical equipment in lying areas, including partitions [43], tie-rails [44], and other solutions providing lying space [45].

Actions undertaken by farmers to improve the comfort of dairy cattle in barns and their surroundings are consistent with the challenges and opportunities presented by technology and the development of intelligent tools in the modern dairy system [46]. Farmer-driven, low-tech innovations can complement and, in some contexts, compensate for commercial sensor-based tools aimed at improving animal comfort and welfare. Sensor-based tools are among the important sources of applications for innovative cattle production technologies, as exemplified by the benefits achieved in animal health diagnostics [47].

The key to implementing innovations in on-farm dairy production research lies in the use of modern tools, including artificial intelligence, to extract and analyze large amounts of data, particularly those related to the behavioral characteristics of dairy cows. Digital and AI-assisted tools can improve the synthesis and reproducibility of data in research on livestock housing and animal welfare [48]. Research on livestock housing and animal welfare has also led to the development of the concept of dairy barns [49], where sustainable herd management is pursued in various areas of the barns and their surroundings, as well as in animal-friendly conditions that facilitate herd functioning [50].

Dairy cattle welfare research requires a specialized approach and the development of appropriately tailored research tools and concepts. In our research study, we did not directly measure animal welfare outcomes (e.g., behavior, health, productivity) in barns with non-standard solutions; therefore, we cannot conclude that the identified technical solutions in the barns improved welfare, only that they were intended to improve animal comfort. It was also not our intention in this research study to interpret the S_ns_ index (index of non-standard solutions in livestock buildings) as a standalone indicator of animal welfare; this proposed index may have complementary functions, serving as supporting information to be considered alongside existing and accepted animal welfare measures.

One limitation of conducting a research study within the scope presented here is the potential lack of sufficient knowledge and practical experience to identify non-standard technical solutions in livestock facilities, as this ability is a result of personal experience and collaboration with farmers who want to demonstrate their ideas for improving technical infrastructure in barns. The above-mentioned personal experience can be considered professional experience; one of the co-authors of this research study has worked for over 20 years at a company specializing in the construction and furnishing of cattle livestock facilities with specialized equipment and metal structures installed in various areas of the barn, which significantly aided in identifying non-standard technical solutions in barns and the reliable collection of research data. The co-author’s extensive professional experience was an asset in recognizing and interpreting non-standard technical solutions; however, to avoid introducing subjective bias, this effect was mitigated during our study by discussing and evaluating non-standard technical solutions with the second co-author and the farmers on each farm.

## 5. Conclusions

At least one non-standard technical solution was identified in each barn with a tie-stall housing system. In the tie-stall barns, the highest number of non-standard technical solutions was found in the feeding area, while in the loose-housing barns, the highest number was found in the social area.

The index of non-standard solutions in livestock buildings (S_ns_) was higher in tie-stall barns (mean S_ns_ 0.0042 [1/m^2^]) than in loose-housing barns (mean S_ns_ 0.0008 [1/m^2^]).

The study results demonstrate farmers’ initiatives to improve comfort, although the direct effects on cow welfare were not measured.

Knowledge about non-standard technical solutions in barns can be applied in practice by various stakeholders as inspiration for barn modernization, advisory materials, or a starting point for targeted welfare research.

## Figures and Tables

**Figure 1 animals-16-00064-f001:**
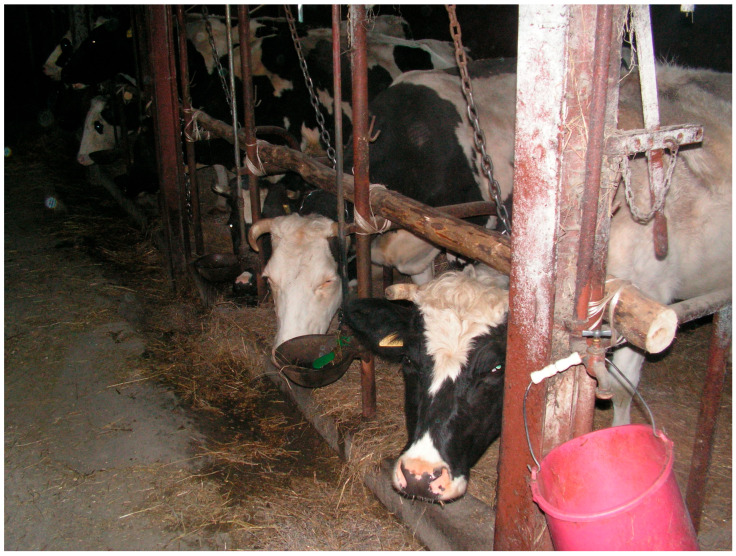
Example of a non-standard solution in the feeding area of a barn with a tie-stall housing system: a horizontal wooden pole in the area dividing the lying stall and the feeding alley (Table 2).

**Figure 2 animals-16-00064-f002:**
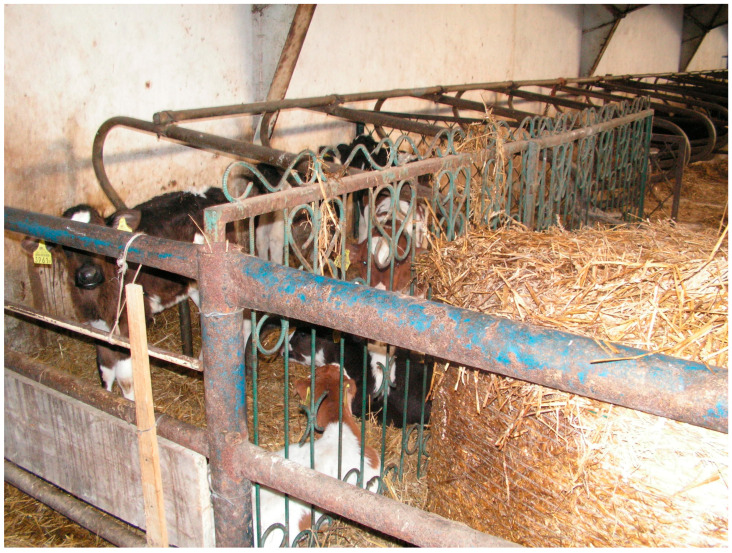
Example of a non-standard solution in the lying area of a barn with a loose-housing system: metal fences from the property, used as partitions for calf pens (Table 3).

**Figure 3 animals-16-00064-f003:**
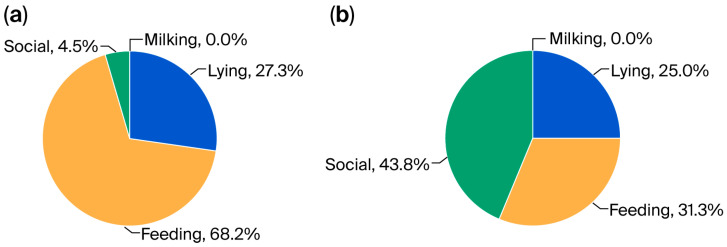
Percentage share (in the total number of identified cases) of non-standard solutions in individual zones (lying, feeding, social, and milking) in barns with (**a**) tie-stall and (**b**) loose-housing systems; the basis for calculating the percentage share was the total number of non-standard solutions, which amounted to 22 and 16 solutions in the barns with the tie-stall and loose-housing systems, respectively.

**Figure 4 animals-16-00064-f004:**
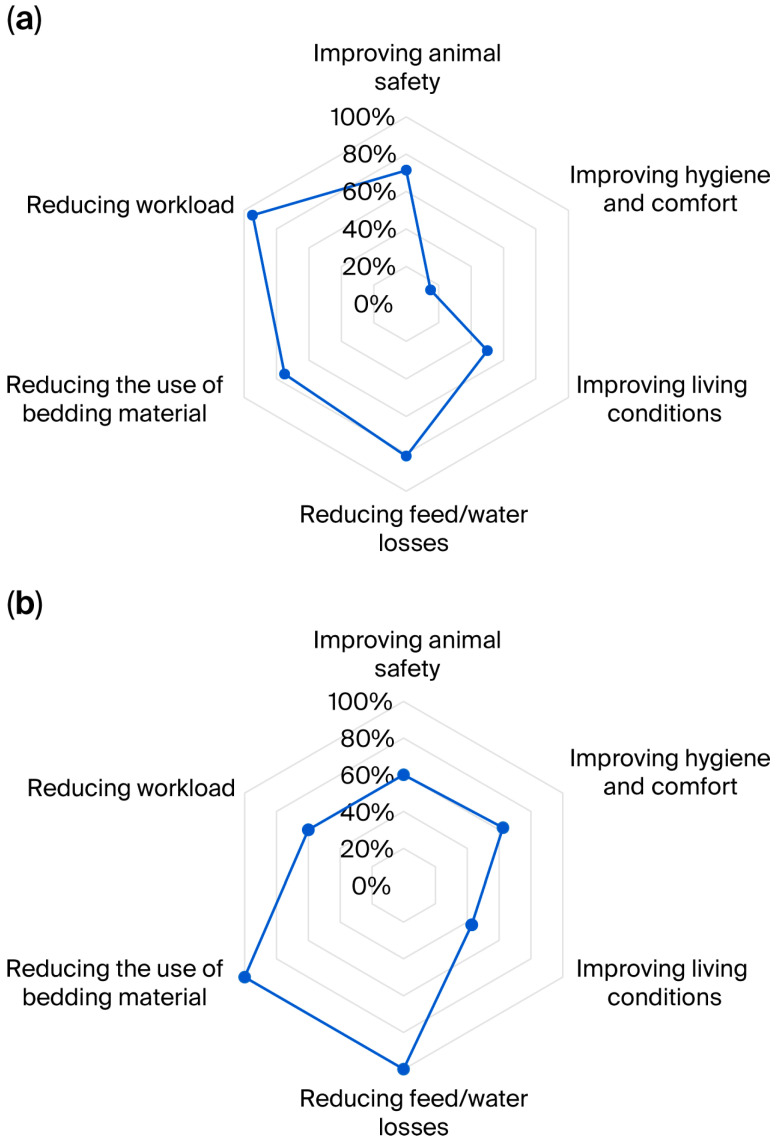
Percentage values representing the proportion of “Yes” responses to all responses (Yes + No) in each evaluation category, in barns with (**a**) tie-stall and (**b**) loose-housing systems; each non-standard solution can contribute up to six ratings (one per category).

**Table 1 animals-16-00064-t001:** Summary of data characterizing the dairy farms covered in this research study.

Data Description	Type of Housing System in the Barn	
	Tie-Stall	Loose-Housing
herd size (ranges)	10–65 cows	35–210 cows
bedding types	rubber mat—1 farm straw—18 farms	rubber mat—9 free-stall barns shallow litter (straw)—10 free-stall barns deep litter (straw)—2 barns
milking systems	bucket milking system—6 farms pipeline milking system—13 farms	milking parlor—18 farms automatic milking system—3 farms

**Table 2 animals-16-00064-t002:** Zones in barns with tie-stall housing systems, and the non-standard solutions identified there.

Zones in Barns with Tie-Stall Housing Systems	Non-Standard Solution	Number of Farms with the Given Case	Expected Effect of the Implemented Solution	Was the Intended Effect Achieved?
Lying	Individual partitions designed by a farmer	3	Improving hygiene and comfort	Yes (×3) *
			Reducing the use of bedding material	Yes (×3)
			Improving animal safety	Yes (×3)
			Reducing workload	Yes (×1)
	Individual wooden partitions designed by a farmer in a separate area	1	Improving living conditions	Yes (×1)
			Improving animal safety	Yes (×1)
	I-beams as vertical columns with a partition function	2	Reducing feed losses	No (×1)
			Reducing workload	Yes (×2)
			Improving living conditions	No (×1)
			Improving hygiene and comfort	No (×1)
			Improving animal safety	No (×1)
Feeding	A horizontal metal pipe at the point dividing the lying stall and the feeding alley	10	Reducing feed losses	Yes (×10)
			Reducing workload	Yes (×10)
			Improving hygiene and comfort	No (×10)
	A horizontal wooden pole in the area dividing the lying stall and the feeding alley	2	Reducing feed losses	Yes (×2)
			Reducing workload	Yes (×2)
			Improving hygiene and comfort	No (×2)
	A metal chain in the area dividing the lying stall and the feeding alley	1	Reducing feed losses	No (×1)
			Reducing workload	Yes (×1)
			Improving hygiene and comfort	No (×1)
			Improving animal safety	No (×1)
	A metal pipe and a wooden pole in the area dividing the lying stall and the feeding alley	1	Reducing feed losses	Yes (×1)
			Reducing workload	Yes (×1)
			Improving hygiene and comfort	No (×1)
	Taking water (by cows) from feed troughs	1	Reducing water losses	No (×1)
			Reducing workload	No (×1)
			Improving hygiene and comfort	No (×1)
Social	Closing the social passage to create an additional lying stall	1	Improving hygiene and comfort	No (×1)
			Reducing the use of bedding material	No (×1)
			Improving animal safety	Yes (×1)
			Reducing workload	Yes (×1)
Milking	Non-standard solutions have not been identified			

* The entry (×3) means that, in three barns, the planned effect was achieved (Yes) or not achieved (No); similar explanations apply to the entries (×1) and others. In other words, “Yes (×3)” indicates the number of dairy farms where this solution achieved the intended effect.

**Table 3 animals-16-00064-t003:** Zones in barns with loose-housing systems, and the non-standard solutions identified there.

Zones in Barns with Loose-Housing Systems	Non-Standard Solution	Number of Farms with a Given Case	Expected Effect of the Implemented Solution	Was the Intended Effect Achieved?
Lying	Boards under the lower mounting pipe at the front of the lying stall	1	Improving hygiene and comfort	Yes (×1) *
			Reducing the use of bedding material	Yes (×1)
			Improving animal safety	Yes (×1)
			Reducing workload	Yes (×1)
	Metal fences from the property used as partitions for calf pens	1	Improving hygiene and comfort	No (×1)
			Improving animal safety	No (×1)
			Reducing workload	No (×1)
	Wooden brackets for the lower mounting pipes of the partitions	1	Improving living conditions	Yes (×1)
			Reducing workload	No (×1)
			Improving animal safety	Yes (×1)
	Wooden gates for the passage of animals, dividing the lying area into individual technological groups	1	Improving living conditions	Yes (×1)
			Reducing workload	No (×1)
			Improving animal safety	Yes (×1)
Feeding	Divided tank of the water bin	1	Improving hygiene and comfort	Yes (×1)
			Reducing water losses	Yes (×1)
			Improving animal safety	No (×1)
			Reducing workload	Yes (×1)
	Using a board between the feed ladder and the concrete wall	1	Improving hygiene and comfort	No (×1)
			Reduction in feed losses	Yes (×1)
			Improving animal safety	Yes (×1)
			Reducing workload	Yes (×1)
	Individually modified feed ladders	1	Improving hygiene and comfort	No (×1)
			Reducing feed losses	Yes (×1)
			Improving animal safety	Yes (×1)
			Reducing workload	No (×1)
	Metal mesh mounted above the feed ladder	2	Reducing feed losses	Yes (×2)
			Improving hygiene and comfort	Yes (×2)
			Reducing workload	Yes (×2)
Social	Metal barrier in the social corridor at the entrance to the milking parlor	2	Reducing workload	Yes (×2)
			Improving living conditions	No (×2)
	A wooden barrier at the edge of the row of lying stalls, separating the social corridor from the passage corridor	1	Improving living conditions	Yes (×1)
			Reducing workload	Yes (×1)
			Reducing the use of bedding material	Yes (×1)
	An additional board at the water bin, at the edge of the row of lying stalls	1	Reducing workload	No (×1)
			Reducing the use of bedding material	Yes (×1)
			Improving animal safety	No (×1)
	Individually modified concrete walls separating the lying modules (sectors)	1	Improving hygiene and comfort	Yes (×1)
			Improving animal safety	Yes (×1)
			Reducing workload	Yes (×1)
	Mounting salt licks on wooden stands	1	Improving living conditions	No (×1)
	Covering the side ventilation slots with hay bags	1	Improving living conditions	No (×1)
			Reducing workload	No (×1)
			Improving animal safety	No (×1)
Milking	Non-standard solutions have not been identified			

* The entry (×1) means that, in one barn, the planned effect was achieved (Yes) or not achieved (No); similar explanations apply to the entries (×2).

## Data Availability

The original contributions presented in this study are included in the article. Further inquiries can be directed to the corresponding author.

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
