# Peer review of "Non-Standard Technical Solutions in Polish Tie-Stall and Loose-Housing Barns: Farmer Initiatives to Improve the Comfort of Dairy Cattle"

_animals, 2025, doi:10.3390/ani16010064_

Round 1

Reviewer 1 Report

Comments and Suggestions for Authors

The Material and Methods chapter is missing the subchapter 2.3. Statistical evaluation.

In the presentation of results, some "non-standard technical solutions" could also be described with a photo. For example, individually modified concrete walls separating the lying modules (sectors); individually modified feed ladders.

When assessing cow welfare, certain blood parameters are also typically determined. Think about it.

The discussion of the results is weak. Are lines 328-337 really relevant to this study?

Author Response

Dear Reviewer,

Thank you for reviewing the article, providing critical comments, and suggesting changes/additions.

The Material and Methods chapter is missing the subchapter 2.3. Statistical evaluation.

At the end of the Materials and Methods chapter, we have added a subsection (2.3) on statistical analyses.

In the presentation of results, some "non-standard technical solutions" could also be described with a photo. For example, individually modified concrete walls separating the lying modules (sectors); individually modified feed ladders.

In the article, we included two photos of examples of non-standard technical solutions that were identified in the visited barns.

When assessing cow welfare, certain blood parameters are also typically determined. Think about it.

We did not directly assess the blood parameters of dairy cattle on farms; therefore, we concluded that we would not expand on the issue of links between blood parameters and animal welfare in our analysis.

The discussion of the results is weak. Are lines 328-337 really relevant to this study?

In the Results and Discussion chapters, we expanded the discussion of research conducted on dairy farms. As part of this expanded discussion, we increased the number of cited articles. We attempted to present various aspects of comparisons between barns with two cattle housing systems, not only regarding the livestock facilities but also the animals housed within them. Therefore, the text includes information on the parameters characterizing cows in tie-stall and loose-housing barns.

Reviewer 2 Report

Comments and Suggestions for Authors

The manuscript presents an observational study on 40 Polish dairy farms (19 tie-stall, 21 loose housing) aimed at documenting “non-standard technical solutions” introduced by farmers in different barn zones (lying, feeding, social, milking). These solutions are classified by location and intended aim (e.g., reducing workload, limiting losses, improving safety/comfort), and the authors also report whether farmers felt that the expected effect had been achieved. An index (Sns = number of non-standard solutions / usable floor area) is proposed to compare barns and housing systems, showing a higher density of non-standard solutions in tie-stall barns. The Discussion links these findings to existing literature on tie-stall vs loose housing, welfare, and management.
In my opinion, the topic is original and practically relevant, but several methodological aspects (farm selection, operational definition of “non-standard technical solutions,” data collection procedures) need clarification for full transparency and replicability. The connection between results and welfare implications could also be sharpened, and the study’s limitations made more explicit. For these reasons, I recommend major revision.

  1. Title and overall framing

Lines 1-4:
I suggest the authors consider adding the housing context and/or geographical scope (e.g., “in Polish tie-stall and loose-housing barns”) to clarify that the study is region-specific and based on descriptive, cross-sectional observations.

  1. Simple Summary

Lines 9-18:
I suggest the authors briefly quantify the main findings (e.g., number of farms with at least one non-standard solution, predominance in tie-stall barns, most-affected barn zones) so that non-specialist readers immediately understand the scale and distribution of the observations.

Lines 14-18:
Please consider clarifying that the study is descriptive and that the authors did not directly measure animal comfort or welfare outcomes, but instead documented farmer-implemented solutions and their perceived effects. In my opinion, this would avoid any risk of overinterpreting “improved comfort” as a measured outcome.

  1. Abstract

Lines 19-33:
I suggest the authors make the sampling strategy more transparent by specifying that farms were purposively selected based on prior knowledge or expectation of non-standard solutions (as detailed later in the Methods). This is important for readers to correctly judge the generalizability of the findings.

Lines 25-29:
I suggest the authors briefly introduce the Sns index in the abstract and report at least a simple comparison between tie-stall and loose-housing barns (e.g., means or range), as this index is presented as a central methodological contribution.

Lines 30-32:
Please consider rephrasing the sentence “More farmers’ technical ideas for improving cattle comfort were identified…” to make clear that this statement is based on the count of non-standard technical solutions, not on direct welfare assessments, so as not to imply causal welfare effects that were not measured.

Lines 31-33:
In my opinion, the utilitarian aim of thed work could be expressed more concretely. I suggest the authors briefly indicate how different stakeholders might use this knowledge (e.g., as inspiration for barn retrofits, training material for advisors, or a starting point for targeted welfare research).

  1. Introduction

Lines 36-56:
I think the introduction gives a good general background, but I suggest the authors more sharply define the research gap: despite substantial work on standard barn designs and commercial equipment, farmer-initiated, “non-standard” modifications are rarely described in a systematic way.

Lines 58-72:
Please consider streamlining this section by more directly linking specific technical factors (stall dimensions, floor type, manures removal system, etc.) to the practical need for farmers to adapt or modify standard solutions.

Lines 73-81:
In my opinion, the shift from welfare concepts to farmer input is appropriate, but I suggest the authors explicitly acknowledge that farmers’ views on “comfort” may differ from scientific welfare indicators and that this study focuses on documenting their technical responses, not on validating welfare outcomes.

Lines 94-96:
I suggest adding one or two concrete examples of “non-standard solutions” (for instance, modified partitions, altered feed barriers, or customized water access points) to help readers visualize what is meant by this key concept.

Lines 97-100:
In my opinion, the brief description of the study design here overlaps with what is later presented in the Materials and Methods Please consider condensing this part or explicitly referring to section 2 to avoid redundancy.

  1. Materials and Methods

5.1. General characteristics of the visited livestock facilities

Lines 101-112:
I think the description of regions, number of farms, and housing systems is clear, but I suggest the authors explicitly state that this was a purposive sample of farms with anticipated non-standard solutions (based on prior contacts and knowledge), rather than a random or representative sample of all farms in the area. This clarification is crucial for transparency.

Lines 113-121:
Please consider summarizing herd size ranges, bedding types, and milking systems in a table contrasting tie-stall and loose-housing farms. In my opinion, this would make it easier for readers to compare the structural characteristics of the two groups.

Lines 115-119:
I suggest the authors clarify whether bedding type distribution differed between tie-stall and loose-housing farms in a way that might influence the probability of implementing non-standard solutions (e.g., more flexibility in deep-bedded systems).

5.2. Approach to research data collection

Lines 123-130:
In my opinion, the description of farm selection via “prior collaboration and professional responsibilities” is a key methodological element and deserves more detail. I suggest the authors indicate: (i) how many candidate farms were initially contacted; (ii) whether any refused to participate; and (iii) whether prior evidence or expectationd of at least one non-standard solution was required for inclusion, or whether some farms were visited without this condition. This would help to assess possible selection bias.

Lines 131-133:
I think it would be useful if the authors specified how long a typical farm visit lasted and whether visits were carried out in comparable periods (season, time of day), as these factors may influence use of outdoor areas and the visibility of certain solutions.

Lines 134-145:
In my opinion, it is important to clarify whether a standardized inspection protocol was used (e.g., a checklist of zones and items) and whether both fauthors always participated in each visit or whether some visits were done by a single observer. This information would strengthen the replicability of the approach.

Lines 140-145:
I suggest the authors clarify whether the four zones considered (feeding, lying, milking, social areas) covered all possible locations where non-standard solutions might be present (e.g., calf pens, transition corridors, waiting areas). If some areas (such as dry cow facilities or separate calf housing) were not included, this should be stated clearly as a limitation either here or in the Discussion.

Lines 146-153:
In my opinion, the description of data recording is useful, but I suggest the authors explicitly indicate whether they applied any formal coding scheme or decision rules to define what counted as a “non-standard solution” (for example, minimum degree of modification, exclusion of simple repairs, etc.).

Lines 151-153:
Please consider clarifying whether farmers’ comments about the origin and perceived effects of the solutions were taken down verbatim, summarized in field notes, or later grouped into categories. In my opinion, a short sentence on how verbal information rwas handled would improve methodological transparency.

Lines 154-160:
I suggest the authors define more precisely what was included in “usable floor space” for the Sns index. For instance, did they count only the areas accessible to cows (lying + feeding + social + milking zones), or also service alleys, technical rooms, or calf pens?

Lines 155-160:
In my opinion, the stated measurement accuracy (1 mm, 1.5 mm) seems unnecessarily detailed compared with the scale of barn dimensions. I suggest eithfer briefly justifying why this level of precision is relevant or simplifying the statement (e.g., “standard tape and laser measuring devices with sub-centimeter accuracy were used”).

Lines 161-171:
I think the Sns index is a potentially useful concept, but I suggest the authors explain more clearly why they chose usable floor area as the denominator, and not—for example—the number of cows, cow places, or pens.
Please consider also acknowledging that Sns implicitly treats all non-standard solutions as equivalent, regardless of complexity or presumed impact. In my opinion, a brief note on this simplifying assumption (in Methods or in the Limitations) would be helpful.

  1. Results

Lines 172-182:
I suggest the authors explicitly state how many total non-standard solutions were identified in the 19 tie-stall barns and how they were distributed among lying, feeding, social, and milking areas, in addition to what is shown in the table.

Table 1 - “Was the expected effect achieved?” column:
I think it would help readers if the caption explained more clearly that “Yes (×3)” indicates the number of farms in which that solution was reported as having achieved its intended effect. Please consider adding a short explanatory sentence in the caption.

Lines 191-199:
I suggest the authors link the description of farmers’ motivations back to the categories utsed in Tables 1 and 2 (e.g., indicating how many times each motivation category occurred). In my opinion, this would make the text more quantitative and anchored to the tables.

Lines 200-203:
I think it would be useful to explicitly state how many farms (8 out of 40) had no non-standard solutions and to indicate whether these were mainly tie-stall or loose-housing farms.

Lines 204-217:
Please consider adding absolute nummbers alongside percentages when describing the distribution of solutions across zones (for example, “21 farms with feeding-area solutions of which X were tie-stall and Y were loose-housing”).

Figure 1 (Lines 230-232):
I think the caption could be slightly expanded to mention the total number of non-standard solutions included for tie-stall and loose-housing barns, so that the reader can better interpret the percentages.

Lines 233-242:
Please consider stating how many total “effect evaluations” (Yes/No) were available for each category and housing system. In my opinion, indicating these denominators would add clarity to the interpretation of Figure 2.

Figure 2 (Lines 283-285):
I suggest the authors revise the caption to specify that each non-standard solution can contribute up to six ratings (one per category) andd that the percentages shown correspond to the proportion of “Yes” responses out of all responses (Yes + No) in each category.

Lines 287-293:
In my opinion, reporting only the range of Sns values is informative but not sufficient. I suggest also presenting mean, median, and possibly interquartile ranges of Sns for tie-stall vs loose-housing barns, and, if the authors deem it appropriate, performing a simple non-parametric test (e.g., Mann-Whitney U) to examine whether differences between the two housing systems are statistically supported, even within the largely descriptive framework of the study.

  1. Discussion

Lines 294-303:
I think the Discussion would be easier to follow if it started with a short recap of the main empirical findings (e.g., prevalence and distribution of non-standard solutions predominance in feeding areas, higher Sns in tie-stall barns) before moving to the broader literature.

Lines 304-347:
Please consider tightening this section and more often linking the cited literature on tie-stall vs loose housing back to the specific result that more non-standard solutions were found in tie-stall barns. In my opinion, it would be useful to discuss possible reasons why farmers in tie-stall systems may feel a stronger need to adapt or modify the environment.

Lines 304-315:
In my opinion, the Discussion could also benefit from referencing this paper (doi:10.2174/0118743315410860250914045935), which discusses the challenges and prospects of wearable technology in dairy cows. This citation would broaden the perspective on how farmer-led, low-tech innovations may complement or in some contexts compensate for commercial sensor-based tools aimed at improving comfort and welfare.

Lines 315-324:
I suggest the authors explicitly state that they did not directly measure animal welfare outcomes (e.g., behavior, health, productivity) in the barns with non-standard solutions, and therefore cannot conclude that the solutions improved welfare, only that they were intended to improve comfort. In my opinion, this clarification would align the interpretation with the study design.

Lines 348-354:
In my opinion, the connection between structural assessment and welfare is appropriate, but I suggest the authors clearly present the Sns index as a complementary indicator that should be considered alongside established welfare metrics, rather than as a standalone proxy for welfare.

Lines 355-362:
I think the paragraph on farmers’ perceptions and personalities is interesting, but I suggest the authors emphasise that these aspects were not directly assessed in the present study and are presented as interpretive context and directions for future work (e.g., investigating links between farmer attitudes and propensity to implement non-standard solutions).
Please consider also citing this manuscript (2025; doi:10.3168/jds.2025-26385), which uses AI for behavioral data extraction in dairy cows. In my opinion, this reference would strengthen the argument about digital and AI-assisted tools for improving data synthesis and reproducibility in livestock housing and welfare research.

Lines 368-373:
I suggest reframing this section to underline that the co-author’s long-term professional experience is a strength for recognizing and interpreting non-standard technical solutions, but that it could also introduce subjective bias. Please consider briefly indicating how this potential bias was mitigated (for example, through joinvt evaluation of solutions or predefined classification criteria).

  1. Conclusions

Lines 374-378:
In my opinion, the conclusions are clear but could be slightly expanded. I suggest adding one or two sentences to: (i) restate the key empirical findings (e.g., higher Sns in tie-stall barns, main barn zones where solutions occurred), and (ii) indicate at least one concrete implication for practice or further research (such as using the documented solutions in advisory materials, or selecting some of them for targeted welfare evaluation).
Please also consider avoiding wording that might be read as evidence of actual welfare improvement (e.g., “improving cattle comfort”) and instead emphasize that the study documented farmer initiatives aimed at improving comfort, without directly measuring cows’ welfare outcomes.

  1. Other sections

Conflicts of Interest:
I suggest the authors clarify whether the second author’s affiliation with a metal structures company could be perceived as a potential conflict of indterest, particularly if similar solutions might be commercially available. In my opinion, a short note stating that the company does not commercially exploit the specific solutions documented in this study (if this is accurate) would help avoid any misunderstanding.

  1. References

I suggest the authors carefully check the journal’s reference style, and ensure consistent formatting of DOIs and other reference elements, as there are minor inconsistencies.

Author Response

Dear Reviewer,

Thank you for reviewing the article, providing critical comments, and suggesting changes/additions. I am very grateful for your review and the time you took to provide all your detailed comments.

  1. Title and overall framing

Lines 1-4:
I suggest the authors consider adding the housing context and/or geographical scope (e.g., “in Polish tie-stall and loose-housing barns”) to clarify that the study is region-specific and based on descriptive, cross-sectional observations.

The title of the article has been revised to include more details, such as the cattle housing systems and the study location.

  1. Simple Summary

Lines 9-18:
I suggest the authors briefly quantify the main findings (e.g., number of farms with at least one non-standard solution, predominance in tie-stall barns, most-affected barn zones) so that non-specialist readers immediately understand the scale and distribution of the observations.

We have added suggested study results at the end of the Simple Summary.

Lines 14-18:
Please consider clarifying that the study is descriptive and that the authors did not directly measure animal comfort or welfare outcomes, but instead documented farmer-implemented solutions and their perceived effects. In my opinion, this would avoid any risk of overinterpreting “improved comfort” as a measured outcome.

In the final part of the Simple Summary, we also included an explanation that the study was descriptive in nature and that we did not directly measure animal comfort/welfare, but documented the solutions implemented by farmers and their opinions.

  1. Abstract

Lines 19-33:
I suggest the authors make the sampling strategy more transparent by specifying that farms were purposively selected based on prior knowledge or expectation of non-standard solutions (as detailed later in the Methods). This is important for readers to correctly judge the generalizability of the findings.

In the Abstract, we have added suggested information regarding strategies for selecting dairy farms for the research study. By the way, we shortened the sentence for the purpose of the research study.

Lines 25-29:
I suggest the authors briefly introduce the Sns index in the abstract and report at least a simple comparison between tie-stall and loose-housing barns (e.g., means or range), as this index is presented as a central methodological contribution.

In the Abstract, we have added suggested sentences regarding the calculation of the proposed Sns index and the differences in its value (with the range given) for barns with tie-stall and loose-housing systems.

Lines 30-32:
Please consider rephrasing the sentence “More farmers’ technical ideas for improving cattle comfort were identified…” to make clear that this statement is based on the count of non-standard technical solutions, not on direct welfare assessments, so as not to imply causal welfare effects that were not measured.

We removed the sentence and rewrote it in its place in such a way that it did not indicate a direct welfare assessment and did not imply causal effects of the study in the area of animal welfare.

Lines 31-33:
In my opinion, the utilitarian aim of the work could be expressed more concretely. I suggest the authors briefly indicate how different stakeholders might use this knowledge (e.g., as inspiration for barn retrofits, training material for advisors, or a starting point for targeted welfare research).

We have removed the last sentence from the Abstract to more precisely present the possibilities for potential stakeholders to use knowledge about non-standard technical solutions in barns in a new sentence.

  1. Introduction

Lines 36-56:
I think the introduction gives a good general background, but I suggest the authors more sharply define the research gap: despite substantial work on standard barn designs and commercial equipment, farmer-initiated, “non-standard” modifications are rarely described in a systematic way.

As suggested, we have defined the research gap more precisely, as presented at the end of the Introduction, because the concept of non-standard technical solutions was not introduced in the initial paragraphs.  

Lines 58-72:
Please consider streamlining this section by more directly linking specific technical factors (stall dimensions, floor type, manures removal system, etc.) to the practical need for farmers to adapt or modify standard solutions.

Following suggestions, we have supplemented this section of the Introduction with details regarding farmers' motivations/needs to improve, modify, or adapt technical equipment in barns. We have included additional citations in this section to reinforce the issues discussed regarding farmers, their awareness, and motivation to modify standard solutions.

Lines 73-81:
In my opinion, the shift from welfare concepts to farmer input is appropriate, but I suggest the authors explicitly acknowledge that farmers’ views on “comfort” may differ from scientific welfare indicators and that this study focuses on documenting their technical responses, not on validating welfare outcomes.

We have supplemented the paragraph in question with suggested issues related to the interpretation of research approaches involving farmers and the analysis of observational results.

Lines 94-96:
I suggest adding one or two concrete examples of “non-standard solutions” (for instance, modified partitions, altered feed barriers, or customized water access points) to help readers visualize what is meant by this key concept.

We have added examples of non-standard technical solutions that can be found in barns in the final paragraph of the Introduction.

Lines 97-100:
In my opinion, the brief description of the study design here overlaps with what is later presented in the Materials and Methods Please consider condensing this part or explicitly referring to section 2 to avoid redundancy.

We have condensed this paragraph into a single sentence to avoid repeating information.

  1. Materials and Methods

5.1. General characteristics of the visited livestock facilities

Lines 101-112:
I think the description of regions, number of farms, and housing systems is clear, but I suggest the authors explicitly state that this was a purposive sample of farms with anticipated non-standard solutions (based on prior contacts and knowledge), rather than a random or representative sample of all farms in the area. This clarification is crucial for transparency.

As suggested, we introduced information that the research study included a purposive sample of farms where we expected non-standard technical solutions in barns.

Lines 113-121:
Please consider summarizing herd size ranges, bedding types, and milking systems in a table contrasting tie-stall and loose-housing farms. In my opinion, this would make it easier for readers to compare the structural characteristics of the two groups.

We have included a table (Table 1) summarizing the data characterizing farms with tie-stall and loose-housing systems.

Lines 115-119:
I suggest the authors clarify whether bedding type distribution differed between tie-stall and loose-housing farms in a way that might influence the probability of implementing non-standard solutions (e.g., more flexibility in deep-bedded systems).

We have made the suggested addition in the section of the article below Table 1.

5.2. Approach to research data collection

Lines 123-130:
In my opinion, the description of farm selection via “prior collaboration and professional responsibilities” is a key methodological element and deserves more detail. I suggest the authors indicate: (i) how many candidate farms were initially contacted; (ii) whether any refused to participate; and (iii) whether prior evidence or expectationd of at least one non-standard solution was required for inclusion, or whether some farms were visited without this condition. This would help to assess possible selection bias.

We have provided more detailed information regarding contact with farmers and selection of farms for the research study.

Lines 131-133:
I think it would be useful if the authors specified how long a typical farm visit lasted and whether visits were carried out in comparable periods (season, time of day), as these factors may influence use of outdoor areas and the visibility of certain solutions.

We have added detailed information regarding farm visits and their timing.

Lines 134-145:
In my opinion, it is important to clarify whether a standardized inspection protocol was used (e.g., a checklist of zones and items) and whether both fauthors always participated in each visit or whether some visits were done by a single observer. This information would strengthen the replicability of the approach.

We have added explanatory sentences regarding the protocol prepared for research/observations on the visited farms. We have also added a sentence about the authors' participation in the farm visits.

Lines 140-145:
I suggest the authors clarify whether the four zones considered (feeding, lying, milking, social areas) covered all possible locations where non-standard solutions might be present (e.g., calf pens, transition corridors, waiting areas). If some areas (such as dry cow facilities or separate calf housing) were not included, this should be stated clearly as a limitation either here or in the Discussion.

We have supplemented the relevant paragraph in this part of the article with information about the locations where different groups of dairy cattle were observed.

Lines 146-153:
In my opinion, the description of data recording is useful, but I suggest the authors explicitly indicate whether they applied any formal coding scheme or decision rules to define what counted as a “non-standard solution” (for example, minimum degree of modification, exclusion of simple repairs, etc.).

We have added two sentences that define non-standard solutions and the criteria for their identification in more detail.

Lines 151-153:
Please consider clarifying whether farmers’ comments about the origin and perceived effects of the solutions were taken down verbatim, summarized in field notes, or later grouped into categories. In my opinion, a short sentence on how verbal information was handled would improve methodological transparency.

Details regarding the collection and processing of information from farmers are expanded upon in an additional sentence.

Lines 154-160:
I suggest the authors define more precisely what was included in “usable floor space” for the Sns index. For instance, did they count only the areas accessible to cows (lying + feeding + social + milking zones), or also service alleys, technical rooms, or calf pens?

The exact definition of “usable area” was developed in the last paragraph of the Materials and Methods chapter.

Lines 155-160:
In my opinion, the stated measurement accuracy (1 mm, 1.5 mm) seems unnecessarily detailed compared with the scale of barn dimensions. I suggest eithfer briefly justifying why this level of precision is relevant or simplifying the statement (e.g., “standard tape and laser measuring devices with sub-centimeter accuracy were used”).

An explanation of the accuracy of the measuring instruments (tape measure and laser meter) is given at the end of the paragraph on measuring usable areas in barns.

Lines 161-171:
I think the Sns index is a potentially useful concept, but I suggest the authors explain more clearly why they chose usable floor area as the denominator, and not—for example—the number of cows, cow places, or pens.
Please consider also acknowledging that Sns implicitly treats all non-standard solutions as equivalent, regardless of complexity or presumed impact. In my opinion, a brief note on this simplifying assumption (in Methods or in the Limitations) would be helpful.

In the section of the article where the formula for Sns and its development are presented, we briefly justified the choice of usable area as the reference parameter in calculating the index of non-standard solutions in livestock buildings. We also noted that the proposed calculation index implicitly treats all non-standard solutions as equivalent, regardless of their complexity or assumed impact.

  1. Results

Lines 172-182:
I suggest the authors explicitly state how many total non-standard solutions were identified in the 19 tie-stall barns and how they were distributed among lying, feeding, social, and milking areas, in addition to what is shown in the table.

We introduced the suggested descriptions in the paragraph before Table 2. Additionally, we added a sentence about the connection between the tie-stall milking area and the lying area in the barn, as well as the identification of non-standard solutions that could be clearly assigned to the lying area.

Table 1 - “Was the expected effect achieved?” column:
I think it would help readers if the caption explained more clearly that “Yes (×3)” indicates the number of farms in which that solution was reported as having achieved its intended effect. Please consider adding a short explanatory sentence in the caption.

As suggested, we have added a short sentence in the caption under the given Table explaining the phrase "Yes (×3)" in more detail.

Lines 191-199:
I suggest the authors link the description of farmers’ motivations back to the categories utsed in Tables 1 and 2 (e.g., indicating how many times each motivation category occurred). In my opinion, this would make the text more quantitative and anchored to the tables.

We wrote a separate paragraph summarizing how many times each motivation was mentioned by farmers, separately in tie-stall and loose-housing barns.

Lines 200-203:
I think it would be useful to explicitly state how many farms (8 out of 40) had no non-standard solutions and to indicate whether these were mainly tie-stall or loose-housing farms.

At the end of the indicated paragraph, we have included information on the number of farms and the type of housing systems that utilized no non-standard technical solutions in the barns.

Lines 204-217:
Please consider adding absolute nummbers alongside percentages when describing the distribution of solutions across zones (for example, “21 farms with feeding-area solutions of which X were tie-stall and Y were loose-housing”).

In the revised paragraph, we included the suggested additional information. We also verified the number of dairy farms in each group of analyzed zones.

Figure 1 (Lines 230-232):
I think the caption could be slightly expanded to mention the total number of non-standard solutions included for tie-stall and loose-housing barns, so that the reader can better interpret the percentages.

In the caption of Figure 1, we have added information about the total number of non-standard solutions in barns with tie-stall and loose-housing systems.

Lines 233-242:
Please consider stating how many total “effect evaluations” (Yes/No) were available for each category and housing system. In my opinion, indicating these denominators would add clarity to the interpretation of Figure 2.

At the end of the given paragraph, we have included data on the total number of "effect evaluations" (Yes/No) for each category and housing system.

Figure 2 (Lines 283-285):
I suggest the authors revise the caption to specify that each non-standard solution can contribute up to six ratings (one per category) andd that the percentages shown correspond to the proportion of “Yes” responses out of all responses (Yes + No) in each category.

As suggested, we have changed the caption under Figure 2.

Lines 287-293:
In my opinion, reporting only the range of Sns values is informative but not sufficient. I suggest also presenting mean, median, and possibly interquartile ranges of Sns for tie-stall vs loose-housing barns, and, if the authors deem it appropriate, performing a simple non-parametric test (e.g., Mann-Whitney U) to examine whether differences between the two housing systems are statistically supported, even within the largely descriptive framework of the study.

In the final paragraph of the Results chapter, we provided the mean and median Sns calculated for the tie-stall and loose-housing barns. We also included the results of the nonparametric Mann-Whitney U test, which compared the index of non-standard solution in the barns with the two housing systems.

  1. Discussion

Lines 294-303:
I think the Discussion would be easier to follow if it started with a short recap of the main empirical findings (e.g., prevalence and distribution of non-standard solutions predominance in feeding areas, higher Sns in tie-stall barns) before moving to the broader literature.

As suggested, we have supplemented the initial part of the Discussion with a summary of the research results conducted on dairy farms.

Lines 304-347:
Please consider tightening this section and more often linking the cited literature on tie-stall vs loose housing back to the specific result that more non-standard solutions were found in tie-stall barns. In my opinion, it would be useful to discuss possible reasons why farmers in tie-stall systems may feel a stronger need to adapt or modify the environment.

We have supplemented the discussion to identify factors that may have contributed to the increased use of non-standard solutions in tie-stall barns for dairy cattle. We have also included citations to specialized literature in this context.

Lines 304-315:
In my opinion, the Discussion could also benefit from referencing this paper (doi:10.2174/0118743315410860250914045935), which discusses the challenges and prospects of wearable technology in dairy cows. This citation would broaden the perspective on how farmer-led, low-tech innovations may complement or in some contexts compensate for commercial sensor-based tools aimed at improving comfort and welfare.

We have quoted the given publication in the appropriate context of the sentences.

Lines 315-324:
I suggest the authors explicitly state that they did not directly measure animal welfare outcomes (e.g., behavior, health, productivity) in the barns with non-standard solutions, and therefore cannot conclude that the solutions improved welfare, only that they were intended to improve comfort. In my opinion, this clarification would align the interpretation with the study design.

In one of the final paragraphs of the Discussion, we wrote suggested sentences regarding the essence of interpreting the results of a research study.

Lines 348-354:
In my opinion, the connection between structural assessment and welfare is appropriate, but I suggest the authors clearly present the Sns index as a complementary indicator that should be considered alongside established welfare metrics, rather than as a standalone proxy for welfare.

In one of the final paragraphs of the Discussion, we suggested sentences regarding the essence of interpreting the Sns index.

Lines 355-362:
I think the paragraph on farmers’ perceptions and personalities is interesting, but I suggest the authors emphasise that these aspects were not directly assessed in the present study and are presented as interpretive context and directions for future work (e.g., investigating links between farmer attitudes and propensity to implement non-standard solutions).
Please consider also citing this manuscript (2025; doi:10.3168/jds.2025-26385), which uses AI for behavioral data extraction in dairy cows. In my opinion, this reference would strengthen the argument about digital and AI-assisted tools for improving data synthesis and reproducibility in livestock housing and welfare research.

We have added the suggested sentence that we did not assess the perception and personality of farmers.

We have cited the given publication in the Discussion section.

Lines 368-373:
I suggest reframing this section to underline that the co-author’s long-term professional experience is a strength for recognizing and interpreting non-standard technical solutions, but that it could also introduce subjective bias. Please consider briefly indicating how this potential bias was mitigated (for example, through joinvt evaluation of solutions or predefined classification criteria).

As suggested, we added two sentences explaining how we aimed to achieve objectivity in identifying and evaluating non-standard barn solutions.

  1. Conclusions

Lines 374-378:
In my opinion, the conclusions are clear but could be slightly expanded. I suggest adding one or two sentences to: (i) restate the key empirical findings (e.g., higher Sns in tie-stall barns, main barn zones where solutions occurred), and (ii) indicate at least one concrete implication for practice or further research (such as using the documented solutions in advisory materials, or selecting some of them for targeted welfare evaluation).
Please also consider avoiding wording that might be read as evidence of actual welfare improvement (e.g., “improving cattle comfort”) and instead emphasize that the study documented farmer initiatives aimed at improving comfort, without directly measuring cows’ welfare outcomes.

As suggested, we have changed and supplemented the conclusions generated from the research study.

  1. Other sections

Conflicts of Interest:
I suggest the authors clarify whether the second author’s affiliation with a metal structures company could be perceived as a potential conflict of indterest, particularly if similar solutions might be commercially available. In my opinion, a short note stating that the company does not commercially exploit the specific solutions documented in this study (if this is accurate) would help avoid any misunderstanding.

In the Conflicts of Interest section, we have added the appropriate information confirming that there was no conflict of interest.

  1. References

I suggest the authors carefully check the journal’s reference style, and ensure consistent formatting of DOIs and other reference elements, as there are minor inconsistencies.

We reviewed the journal's citation style and DOI formatting, removing any minor inconsistencies we observed.

Round 2

Reviewer 1 Report

Comments and Suggestions for Authors Of little importance - it can hardly be called scientific research, despite the corrections.

Author Response

Of little importance - it can hardly be called scientific research, despite the corrections.

Answer: Thank you for your feedback on the research and article. 

Reviewer 2 Report

Comments and Suggestions for Authors

the paper improved a lot, I have no further comments

Author Response

The paper improved a lot, I have no further comments.

Answer: Thank you for your final feedback on the revised version of the article.